# Chest Wall Reconstruction Using Titanium Mesh in a Dog with Huge Thoracic Extraskeletal Osteosarcoma

**DOI:** 10.3390/ani14182635

**Published:** 2024-09-11

**Authors:** Woo-June Jung, Ho-Hyun Kwak, Junhyung Kim, Heung-Myong Woo

**Affiliations:** Department of Veterinary Surgery, College of Veterinary Medicine, Kangwon National University, Chuncheon 24341, Republic of Korea

**Keywords:** titanium mesh, chest wall, reconstruction, extraskeletal osteosarcoma, dog

## Abstract

**Simple Summary:**

In this study, we present the case of a dog diagnosed with a large thoracic wall extraskeletal osteosarcoma, which was successfully treated through chest wall reconstruction using a titanium mesh. This approach involved the en bloc resection of multiple ribs and the secure placement of a titanium mesh to reconstruct the chest wall. The dog showed significant improvement post-surgery, with no signs of disease recurrence or metastasis at the 8-month follow-up. This case highlights the effectiveness of titanium mesh in providing structural support and promoting recovery in dogs undergoing extensive chest wall resections due to large tumors.

**Abstract:**

A 6-year-old castrated male mixed dog presented with a rapidly growing mass at the right chest wall two weeks after initial detection. A mesenchymal origin of the malignancy was suspected based on fine-needle aspiration. Computed tomography (CT) revealed that the mass originated from the right chest wall and protruded externally (6.74 × 5.51 × 4.13 cm^3^) and internally (1.82 × 1.69 × 1.50 cm^3^). The patient revisited the hospital because of breathing difficulties. Radiography confirmed pleural effusion, and ultrasonography-guided thoracocentesis was performed. The effusion was hemorrhagic, and microscopic evaluation showed no malignant cells. Before surgery, CT without anesthesia was performed to evaluate the status of the patient. The 7–10th ribs were en bloc resected at a 3-cm margin dorsally and ventrally, and two ribs cranially and caudally from the mass. After recovering the collapsed right middle lobe of the lung due to compression from the internal mass with positive-pressure ventilation, a 3D-printed bone model contoured titanium mesh was tied to each covering rib and surrounding muscles using 2-0 blue nylon and closed routinely. The thoracic cavity was successfully reconstructed, and no flail chest was observed. The patient was histo-pathologically diagnosed with extraskeletal osteosarcoma. A CT scan performed 8 months after surgery showed no evident recurrence, metastasis, or implant failure. This is the first case report of chest wall reconstruction using titanium mesh in a dog. The use of a titanium mesh allows for the reconstruction of extensive chest wall defects, regardless of location, without major postoperative complications.

## 1. Introduction

Osteosarcoma (OSA) is the most common tumor of the canine thoracic wall, followed by chondrosarcomas, fibro-sarcomas, and hemangiosarcoma [1,2,3]. OSA is characterized by the formation of osteoids, which can occur not only in the limbs but also in other areas, such as the ribs, sternum, muscles of the chest wall, and pleura. In addition, OSA originating from the ribs can spread into the lungs and thoracic cavity [4]. Diagnosing OSA requires confirmation through biopsy, which can be performed using techniques, such as fine-needle aspiration or Jamshidi needle biopsy.

Regardless of the tumor type, the recommended approach is surgical resection (en bloc excision) as, due to the poor prognosis associated with conservative management of thoracic wall tumors in dogs and the significant impact of local tumor recurrence on survival time, a more aggressive treatment approach may be warranted [1,5]. However, when the tumor is large or has extensively invaded the thoracic cavity, extended resection may be necessary to ensure that all cancerous tissue is removed, leaving a clear surgical margin. During the surgical excision of a tumor from the chest wall, removing a maximum of six to seven ribs has been reported [6,7]; however, removing more than four ribs can lead to the development of a flail chest [8]. A flail chest occurs when a segment of the rib cage detaches from the rest of the chest wall due to multiple rib fractures or extensive rib resection. This results in a paradoxical movement of the affected chest wall during respiration because the detached segment moves inward during inhalation and outward during exhalation, in contrast to the normal chest wall movement.

Various methods can be used to reconstruct chest wall defects after excision. A musculocutaneous flap using the latissimus dorsi effectively covers defects on the cranial side of the chest wall, whereas diaphragmatic advancement is effective for reconstruction on the caudal side [9,10,11,12]. The chest wall can also be reconstructed using a polypropylene mesh [13,14]. Small thoracic defects can be reconstructed using existing methods. However, applying these methods may be challenging depending on the location and size of the tumor. Additionally, reconstruction using polypropylene mesh can lead to complications, including infection, dehiscence, seroma, and hematoma, introducing challenges in reconstructing large thoracic defects [8,15,16].

Titanium has numerous advantages, including biocompatibility, strength, durability, and light weight [17,18,19,20,21,22,23,24,25]. Therefore, it is widely utilized in various medical applications, such as dental implants, joint prostheses, and bone fixation devices. This is the first report of chest wall reconstruction using a titanium mesh in a dog, demonstrating the applicability of titanium for large defects.

## 2. Case Description

A 6-year-old, 6.7 kg, castrated male mixed breed dog presented to the Kangwon National University Veterinary Teaching Hospital, Chuncheon, Republic of Korea, with a rapidly growing mass around the right chest wall two weeks after initial detection. Physical examination revealed that the mass was firmly adhered to the right side of the chest wall. Vital signs were unremarkable except for a mildly elevated respiratory rate (55/min). Complete blood count (CBC) and serum biochemistry results were normal, except for elevated alkaline phosphatase levels (308 U/L, reference interval 15–127 U/L). Fine needle aspiration and cytology were performed before surgery. The cytological diagnosis indicated a malignant tumor of mesenchymal origin or activated mesenchymal cells. Radiography and computed tomography (CT) were performed to evaluate the tumor size, location, adhesions, and the presence of metastasis to the lymph nodes or other organs. The tumor penetrated the chest wall at the 8th intercostal space (Figure 1A–D). The size of the outer mass was 6.74 × 5.51 × 4.13 cm^3^ (length × width × height), and that of the inner mass was 1.82 × 1.69 × 1.50 cm^3^. CT showed an inner mass that appeared to have invaded the right middle lobe of the lung, with no metastasis to other organs.

Based on the CT scans, a rib model was printed using a 3D printer. The rib model was used to determine the surgical margin and contour of the titanium mesh in preparation for surgery (Figure 2A,B). The surgical plan was en bloc resection of the 7th to 10th ribs with lateral margins set 3 cm from the ventral and dorsal edges of the tumor at two cranial and caudal ribs.

The patient presented to the emergency room 13 days after the first visit, with dyspnea as the chief complaint. Radiographic imaging showed increased radiopacity of the thoracic cavity, particularly in the right lung lobe (Figure 1E,F). Thoracocentesis was performed, and the breathing of the patient improved. The pleural effusion was hemorrhagic, and cytology following aspiration did not reveal any identifiable cells. Considering the patient’s condition, non-contrast CT was performed without anesthesia. Upon comparison with previous scans, a marked increase in calcification of the external mass and suspicion of rupture of the internal mass were observed (Figure 1G,H), leading to the decision to perform emergency surgery.

The dog was premedicated with midazolam 0.3 mg/kg intravenously (IV) (Midazolam, Inj^®^; Bukwang Pharm, Seoul, Republic of Korea). Anesthesia was induced using propofol 6 mg/kg (IV) (Anepol Inj^®^, Hana Pharm, Seoul, Republic of Korea). Cefazoline 22 mg/kg (IV) (Cefazoline Inj^®^, Chongkundang, Republic of Korea), a prophylactic antibiotic, was administered before the operation. Intraoperative analgesia was provided via continuous rate infusion of a combination of fentanyl (0.002 mg/kg/h, Fentanyl Citrate Inj^®^, Hana Pharm, Republic of Korea), lidocaine (1.5 mg/kg/h, Lidocaine Hcl Hydrate Inj. 2%^®^; Daihan, Republic of Korea), and ketamine (0.15 mg/kg/h, Ketamine 50 Inj^®^, Yu-han, Republic of Korea), which is referred to as FLK. The hair was clipped widely. The dog was placed in the left lateral recumbent position (Figure 2C). A 3-cm margin around the mass was delineated using a sterile pen. An incision was made along the indicated border, and the tissue beneath the skin was incised, perpendicular to the designated margins. The latissimus dorsi muscle and the tumor were extensively excised from the chest wall. The tumor was dissected from the surrounding tissues using Bovie (Covidien, Galway, Ireland) and LigaSure (Covidien, Dublin, Ireland) (Figure 2D). A 3-cm margin ventrally and dorsally and two ribs cranially and caudally were observed around the tumor’s attachment to the chest wall, and the 7th to 10th ribs were removed en bloc using a rongeur (Figure 2E). A collapsed right middle lobe was observed, likely due to an inner mass (Figure 2F). Subsequently, recovery was facilitated via positive-pressure ventilation (Figure 2G). Although no cells were observed in the pleural effusion, the thoracic cavity was irrigated with warm saline. A titanium mesh (60 × 70 × 0.6 mm^3^, Veterinary Instrumentation, Sheffield, UK) was positioned to cover the tumor resection site with a 1-cm margin [26]. Subsequently, 2-0 blue nylon was used to suture the mesh along the margins of the ribs without passing through it, and the surrounding muscles were sutured together (Figure 2H). A chest tube was placed before routinely closing the defect. No complications occurred during the surgery.

The constant rate infusion of FLK was maintained as postoperative analgesia for 3 days, and amoxicillin/clavulanate was administered intravenously twice daily. No flail chest was observed, and the patient breathed stably. On the following day, the appetite and vital signs improved. The chest tube was removed on postoperative day two. The patient was discharged from the hospital on postoperative day five. The sutures were removed on post-surgery day 14.

Histopathological examination revealed that the resected mass was an osteoblastic OSA without vascular invasion (Figure 3). The specimen was characterized by a poorly demarcated, unencapsulated, and highly cellular mass that effaced the skeletal muscle and fibroadipose connective tissue. Neoplastic cells with multinucleated giant cells and osseous matrix were also observed. The mitotic count was 12 mitoses per 2.37 square millimeters.

The owner of the dog opted not to pursue chemotherapy, instead preferring periodic follow-up. At the 8 postoperative months follow-up, the overall condition of the patient was generally good, showing no signs of dyspnea or recurrence and no metastasis or evident disease progression on the radiography or CT scans (Figure 4).

## 3. Discussion

OSA is a highly malignant mesenchymal tumor. It is the primary bone tumor in dogs, representing 85% of all documented bone tumors [27,28]. The skeleton is commonly involved and can be used to classify OSA into axial and appendicular forms [27,28,29]. The axial form of OSA refers to its occurrence primarily in the deep bones, including the spine, pelvis, skull, and other central bones. The appendicular form of OSA primarily originates in the long bones of the arms and legs, such as the humerus and femur. These tumors are common and often have the potential to spread to the muscles and surrounding tissues. Among OSAs, the less common form, which does not involve the primary bone tissue, is referred to as extraskeletal OSA and is rare in dogs. Extraskeletal OSA is subcategorized into mammary gland and soft tissue OSA, accounting for 64% and 36% of cases, respectively [30]. Extraskeletal OSA commonly occurs in the mammary glands, gastrointestinal tract, subcutaneous tissues, spleen, urinary tract, liver, skin, muscle, eye, and thyroid gland. Additionally, there have been case reports of its occurrence in the salivary glands, esophagus, omentum, pericardium, liver, caudal vena cava, retroperitoneal space and hair follicle [30,31,32,33,34,35,36,37,38,39,40,41].

Extraskeletal OSA is commonly identified in older individuals [30,41,42,43]. In contrast to skeletal OSAs, they are not associated with large breed dogs [43]. No specific breed or sex preferences were noted [30,41,42,44]. It manifests as rapid growth, displaying irregular mineralized regions alongside hemorrhagic and necrotic areas [44,45]. These formations, with distant metastases, observed in up to 64% of cases, exhibit a high rate of local recurrence and metastasis [41,43,44,45]. However, lung metastasis is relatively infrequent compared with skeletal OSA [30,43]. Lung metastasis typically has a poor prognosis and short post-diagnosis survival time, possibly due to delayed detection [42,43]. Fatality commonly occurs due to local recurrence or euthanasia after diagnosis [30,43]. Diagnosis is usually confirmed using clinical, radiographic, and histological findings [42]. Certain criteria must be met histologically to confirm the diagnosis, including a consistent morphological pattern of sarcomatous tissue, exclusion of the likelihood of a mixed mesenchymal tumor, generation of woven bone or malignant osteoid tissue, high mitotic index, and ruling out of a bone origin [41,44]. In the present case, considering the absence of positive findings on radiographs, lack of association with bone on histopathological examination, and a history of an intramuscular mass extending into the thoracic wall, the diagnosis was suspected to be extraskeletal OSA, even within the context of OSA.

Chest wall reconstruction following rib tumor excision, which includes a latissimus dorsi flap, diaphragm advancement, and the use of polypropylene mesh, has been reported [9,10,11,12,13,14,46]. However, there is no established gold standard method for thoracic wall reconstruction. In the present case, a portion of the latissimus dorsi muscle and four ribs were resected to ensure adequate surgical margins around the tumor. Therefore, a robust structure was required to prevent a flail chest, and a titanium mesh was used to reconstruct the thoracic wall defect. The thoracic cavity was successfully reconstructed without major postoperative complications, including flail chest and titanium mesh failure or migration.

Titanium is safe for medical use with minimal reactivity and toxicity [17,19,21,23,24,25]. It does not corrode or degrade in the body, making it suitable for long-term use. Titanium is incredibly strong and durable, providing excellent resistance to impact and pressure. This ensured that the implanted titanium devices can withstand stresses encountered by the body. Titanium is lightweight, which reduces the burden on the implanted areas and makes it comfortable for patients. Titanium exhibits high physiological stability, allowing it to integrate well with the bone tissue and maintain high biocompatibility. This ensures the long-term stability and integration of titanium implants. Recently, titanium meshes with greater strength than synthetic meshes have been proposed to maintain the same plasticity and adaptability for chest wall defects [23]. A 0.5-mm thick mesh can provide sufficient rigidity to the chest wall, preventing damage to the intrathoracic organs and preserving the integrity of the system because of the inert nature of titanium, resulting in excellent tolerance [23]. We used titanium mesh instead of bone regeneration materials for rib reconstruction. This decision was based on a study in which BMP-2 significantly promoted OSA cell growth and increased tumor cells’ mobility and invasiveness in vitro [47]. Reconstruction following tumor resection does not rule out the risk of the carcinogenic potential associated with substances, such as bone morphogenetic protein-2 (BMP-2). Further research is required to determine the risks involved. Following chest wall reconstruction using a titanium mesh, a fibrous tissue layer called the pseudo-pleura forms within two weeks [48]. The pseudo-pleura is a protective barrier against direct entry into the pleural cavity. However, the reasons for the formation of these layers remain unknown. This could be attributed to an inflammatory reaction secondary to the infection or the presence of the titanium mesh itself, which may have facilitated the formation of granulation tissue.

In appendicular osteosarcoma, elevated levels of alkaline phosphatase (ALP) are strongly linked with shorter overall survival times, reduced event-free survival rates, and the presence of metastasis at the time of diagnosis. In contrast, little is known about the prognosis and prognostic factors specifically related to extraskeletal osteosarcoma. Further research is needed [49]. Tumor rupture can lead to hypovolemic shock, which may result in a poor prognosis. However, there are currently no studies examining the relationship between the rupture of thoracic tumors and prognosis. Further research is needed. This is particularly concerning when the rupture occurs in the thoracic cavity, as it can cause respiratory distress, necessitating immediate intervention [50]. The median survival time (MST) for extraskeletal OSA varies depending on the treatment received and ranges from 26 to 190 days [30,41,51]. Extraskeletal OSAs arising in the subcutaneous tissue or muscle have a longer survival than those occurring in the abdominal organs or omentum [30,41,51]. However, statistically proven data regarding this are yet to be established. This case report represents the first instance in veterinary literature documenting the occurrence of an extraskeletal OSA originating from the chest wall of a dog. In this case, 8 months have passed since the surgery, with no recurrence or distant metastases. Furthermore, the titanium mesh has been maintained without failure or migration since the surgery. However, long-term follow-ups are necessary.

Fractures occurred in the 6th (Figure 4E) and 11th ribs, presumably due to friction with the non-absorbable sutures during breathing. In some cases, excessive rib retraction may lead to rib fractures, although these do not typically require corrective treatment [52]. This requires careful observation, as it can be mistaken for OSA recurrence.

## 4. Conclusions

In conclusion, to the best of our knowledge, this is the first case report of successful chest wall reconstruction using a titanium mesh in a dog without major postoperative complications, including a flail chest. A titanium mesh is a biocompatible implant that can be effectively applied to any thoracic defect regardless of the location and is suitable for extensive chest wall reconstruction.

## Figures and Tables

**Figure 1 animals-14-02635-f001:**
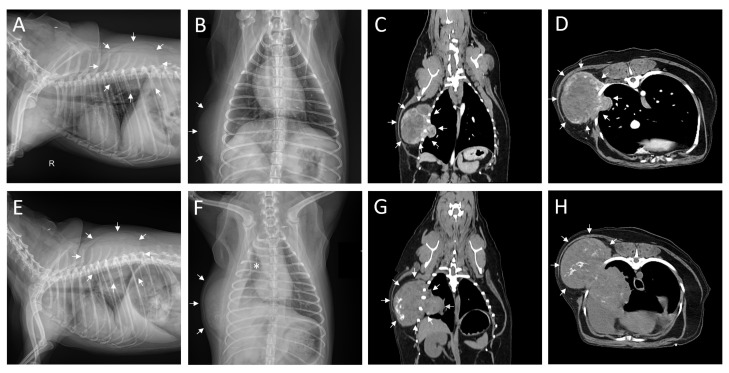
Radiograph and computed tomography images (**A**–**D**, first visit; **E**–**H**, revisit after 13 days) of a 6-year-old mixed breed dog with right chest wall mass. In all figures, white arrows indicate mass. Right-lateral (**A**) and ventro-dorsal (**B**) thoracic radiograph demonstrating a radio-opaque mass occupying the right thoracic wall. Contrast-enhanced computed tomography (CT) images of coronal (**C**) and axial (**D**) planes of the patient. The mass extending externally and internally into the thoracic cavity was observed (**C**,**D**). Right-lateral (**E**) and ventro-dorsal (**F**) thoracic radiographs taken during the revisit when the patient presented with dyspnea as the chief complaint. Calcification within the mass and radio-opaque thoracic cavity was observed with the notable collapse of the right lung lobe (**E**,**F**, asterisk). Preoperative non-contrast CT images of coronal (**G**) and axial (**H**) planes without anesthesia show an increase in the size of the external thoracic mass and, although the exact boundary of the internal thoracic mass was unclear, there was a suspicion of rupture.

**Figure 2 animals-14-02635-f002:**
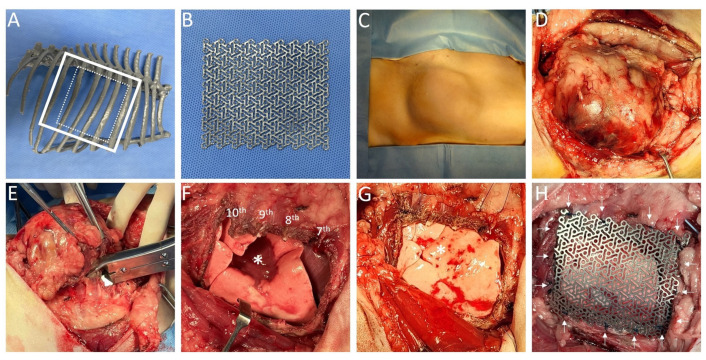
Surgical planning and procedure of 7th to 10th en bloc ribs resection of the mass from the right chest wall and thoracic reconstruction using titanium mesh. In the surgical procedure figures (**C**–**H**), the right is the cranial side, and the left is the caudal side. Based on the CT scans, a rib model was printed using a 3D printer, which was used to determine the resection margin and placement of the contoured titanium mesh (**A**). A titanium mesh implant (70 × 60 × 0.6 mm^3^) was used (**B**). The dog in lateral position (**C**). Incision of the skin with a 3-cm margin from the tumor, exposing the mass (**D**). Resection of the 7th rib from the 10th rib using bone cutter (**E**). The right middle lobe (asterisk), which was being compressed by an internal tumor, was observed after the resection (**F**). The right middle lobe (asterisk) was recovered using positive-pressure ventilation (**G**). The titanium mesh was securely sutured along the margins of the 6th to 11th ribs using 2-0 blue nylon (**H**, white arrow).

**Figure 3 animals-14-02635-f003:**
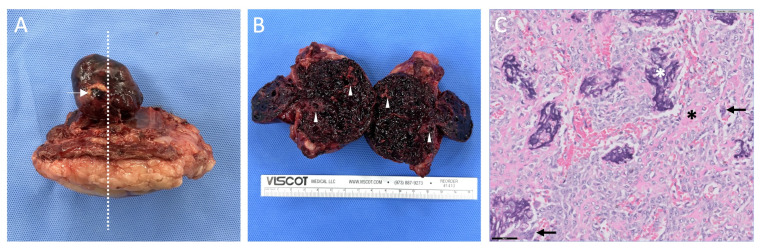
Gross images (**A**,**B**) and histopathological image (**C**) of resected mass. The suspected rupture site in the internal tumor was identified (**A**, arrow), and the tumor was transected between the 8th and 9th rib (**A**, dotted line). Calcifications were observed in external mass (**B**, arrowhead). HE stained neoplastic cells with multinucleated giant cells (**C**, arrow), and dark purple mineralized (**C**, white asterisk) and eosinophilic unmineralized (**C**, black asterisk) osseous matrix at 400× magnification (**C**).

**Figure 4 animals-14-02635-f004:**
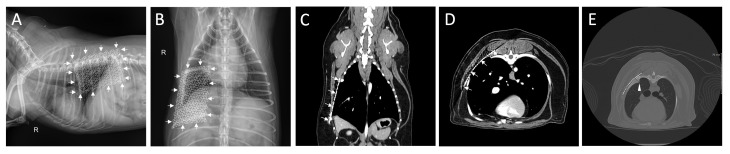
Radiograph (**A**,**B**) and computed tomography images (**C**,**D**) at 8 months post-operation. There was no metastasis or evident progression of disease and migration or failure of titanium mesh (arrow). This image shows a fracture in the 6th rib (**E**, arrowhead), observed on a follow-up CT scan five weeks after surgery. A fracture in the 11th rib was also noted. By the eight-month postoperative follow-up, no significant changes were observed in either rib.

## Data Availability

No new data were created or analyzed in this study. Data sharing is not applicable to this article.

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
