# Peer review of "Chest Wall Reconstruction Using Titanium Mesh in a Dog with Huge Thoracic Extraskeletal Osteosarcoma"

_animals, 2024, doi:10.3390/ani14182635_

Round 1

Reviewer 1 Report

Comments and Suggestions for Authors

Congratulations for the authors to present a case of Chest Wall Reconstruction Using Titanium Mesh in a Dog with extraesk. OSA. The case report is very illustrative with a lot of details for the reader and to understand the approach use of titanium mesh in this localization. 

There are some points that were raised during the reading of the paper which needs attention. 

Abstract:
-Line 18- " 2" for two

- Line 19: please include the size of the mass

- Line 24: What authors mean about "status of the patient" with a CT without anesthesia. Before surgery, how many days after consult it took for that, because looked like the CT was to plan again the surgery process.

Case description

- Line 71- "castrated male mixed breed dog"

- Line 73- "2" for two

- Line 75 - Which Blood analysis were performed? CBC and bioch panel I suposed; Please detailed.

-Line 82- Cm or Cm³? please check it

-Line 88 - Breed dog (legend of the figure 1)

- LIne 94 - Please include an Asterisk to refer the collapse lung.

- Line 109- do not need to abbreviate "Lt."

- Line 148 - Dosage of Amoxi; twice daily

-Line 151- if the FLK was stopped at day 3 posop, and discharge at day 5 postop; which was the analgesic therapy managment for this patient? Furthermore, It was discharged at home with which prescription.

- Line 152. How was the management of the patient at home? Any special care for the titanium mesh? Any minor complication posop?

- Line 229 - bone morphogenetic protein-2 (BMP-2)?

- Line 248- The report developed fracture? Or you are mentioning the literature? Check, there is no reference.

Author Response

Thank you so much for giving opportunity to revise our manuscript “Chest Wall Reconstruction Using Titanium Mesh in a Dog with Huge Thoracic Extraskeletal Osteosarcoma” for MDPI in Animals. We want to extend our appreciation to you and the reviewers for taking the time and effort necessary to provide such insightful guidance. We have carefully considered comments offered by the reviewers. Herein, we explain how we revised the paper based on those comments and recommendations. We look forward to working further with you and the reviewers to move this manuscript closer to publication.

COMMENT 1.

Abstract: - Line 18 " 2" for two

RESPONSE 1.

Thanks for your detail comments. The revised content was reflected in the revised edition. We hope that these revised results will be a sufficient answer and will help to publish our research results in this journal.

Abstract - Line 18 : We changed “2” to “two”.

COMMENT 2.

- Line 19: please include the size of the mass

RESPONSE 2.

Thanks for your detail comments.

- Line 20: We changed “externally and internally” to “externally (6.74 × 5.51 × 4.13 cm³) and internally (1.82 × 1.69 × 1.50 cm³)”

COMMENT 3.

- Line 24: What authors mean about "status of the patient" with a CT without anesthesia. Before surgery, how many days after consult it took for that, because looked like the CT was to plan again the surgery process.

RESPONSE 3.

Thank you for your careful review. The dog was in respiratory distress due to pleural effusion. Although partial removal of the effusion was performed via thoracocentesis, lung function remained compromised. Considering the patient’s condition, a non-anesthetic, non-contrast CT scan was conducted. Additionally, the dog presented to the emergency room 13 days after the initial visit (with some delay due to a weekend and Korean traditional holidays)."

COMMENT 4.

Case description

- Line 71- "castrated male mixed breed dog"

RESPONSE 4.

Thanks for your detail comments.

- Line 73: We changed “castrated male mixed dog” to "castrated male mixed breed dog"

COMMENT 5.

- Line 73- "2" for two

RESPONSE 5.

Thanks for your detail comments.

- Line 75: We changed “2” to "two"

COMMENT 6.

- Line 75 - Which Blood analysis were performed? CBC and bioch panel I suposed; Please detailed.

RESPONSE 6.

Thank you for your detail comments.

- Line 78: We changed “Blood analysis” to “Complete blood count (CBC) and serum biochemistry”

COMMENT 7.

-Line 82- Cm or Cm³? please check it

RESPONSE 7.

Thanks for your detail comments.

- Line 82: We changed “Cm3” to " Cm³"

COMMENT 8.

-Line 88 - Breed dog (legend of the figure 1)

RESPONSE 8.

Thanks for your detail comments.

- Line 90: We changed “mixed dog” to "mixed breed dog"

COMMENT 9.

- LIne 94 - Please include an Asterisk to refer the collapse lung.

RESPONSE 9.

Thanks for your detail comments. We included asterisk on image and legend of Figure 1F.

- Line 94: We changed “notable collapse of the right lung lobe (E, F)” to " notable collapse of the right lung lobe (E, F, asterisk)".

COMMENT 10.

- Line 109- do not need to abbreviate "Lt."

RESPONSE 10.

Thanks for your detail comments.

- Line 109: We deleted "Lt."

COMMENT 11.

- Line 148 - Dosage of Amoxi; twice daily

RESPONSE 11.

Thanks for your detail comments.

- Line 148: We chagend "2 times a day" to "twice daily"

COMMENT 12.

-Line 151- if the FLK was stopped at day 3 posop, and discharge at day 5 postop; which was the analgesic therapy management for this patient? Furthermore, It was discharged at home with which prescription.

RESPONSE 12.

Thank you so much for your careful review. After surgery, we also medicated esomeprazole (Nexium, Inj®; AstraZeneca Korea, Korea) at a does of 1mg/kg IV twice daily and meloxicam (Metacam, Inj®; Boehringer Ingelheim Korea, Korea) at a dose of 0.2 mg/kg IV on the first day of treatment; subsequent doses of 0.1 mg/kg IV once daily upon discharge. After discharge, both medications were transitioned to oral administration and continued until suture removal.

COMMENT 13.

- Line 152. How was the management of the patient at home? Any special care for the titanium mesh? Any minor complication posop?

RESPONSE 13.

Thank you so much for your careful review. Daily visits were required for dressing changes and bandage management. At home, the owner was instructed to restrict the dog’s activity, with no special care needed aside from monitoring the respiratory rate. As a postoperative minor complication, the dog experienced mild pancreatitis, which resolved by the third postoperative day, and there were no complications related to the titanium mesh.

COMMENT 14.

- Line 229 - bone morphogenetic protein-2 (BMP-2)?

RESPONSE 14.

Thank you so much for your careful review. Yes, instead of using bone morphogenetic protein-2 (BMP-2) for rib regeneration, we opted to reconstruct the thoracic cavity using a titanium mesh.

COMMENT 15.

- Line 248- The report developed fracture? Or you are mentioning the literature? Check, there is no reference.

RESPONSE 15.

Thank you so much for your careful review. In some cases, excessive rib retraction may lead to rib fractures, although these do not typically require corrective treatment.

We included CT image of fractured rib at Figure 4. and reference at the pharagraph.

Reviewer 2 Report

Comments and Suggestions for Authors

This is a very nice start of a case report, with some inclusion of case follow up and description of the use of the mesh in the discussion, it will be a great option for future cases.

Title:  Rather than use the colloquial term "huge" please use "large" or "very large" to describe the size of the mass.

Line 44-46:  This sentence suggests that surgery improves survival time and is not palliative as it is in cases of appendicular osteosarcoma.  The introduction or discussion should include this information with references.

Line 44-64:  The reader appreciates this paragraph.  Can the authors provide information about using plates to support chest wall reconstructions when defects >4 ribs are made?  This information does exist in the literature and is likely what readers will recall and to which they will refer.

Line 76:  Does the increase in ALP affect prognosis in cases like this as it does for appendicular osteosarcoma?  Please include this in the discussion of this manuscript.

Line 80:  Please change "synechia" to "adhesions", as most readers understand that synechia refers to the eye.

Line 84:  Please change "and no metastasis" to "with no metastasis"

Figure 1, part G:  Was there any change in the ribs associated with the mass?  Osteolysis, osteoproliferation?  Those changes are important in determining if this is axial osteosarcoma.

Line 100-101:  Thank you for including a very nice description of the intended surgical excision including ribs and cm to be removed and the method of preparing the mesh.

Line 110:  The instrument in Figure 2 part E seems to be a bone cutter or bone cutting forceps.  Please change "rongure" to one of the other terms.

Line 121:  If the mass did rupture, please address this in the discussion and how it may affect survival.  Please include any references as to the conculsions made.

Line 138:  Please change "rongure" to "bone cutting forceps"

Line 141:  Please discuss how saline lavage minimizes the possibility of  metastasis.  If no literature is available to support this statement, please remove it form the portion on "seeding".

Line 144:  Please change "tie" to "suture"

Line 144-146:  Was no omentum used to protect the lungs from the mesh?  This is important and should be included in the discussion as well.

Line 155:  Please change "effaces" to "effaced".

Line 153-157:  Were margins clear of neoplastic cells?  If yes, please provide the measurement of tumor-free tissue obtained.  This is very important to the reader and potentially to the complications of recurrence expected.

Figure 4:  Can images of the CT scan be included that are more appropriate for evaulation of lung for metastasis?  The current images do show the mesh very nicely.  But evaluation of bone, soft tissue, and lung are important for follow up.

Line 190-206:  This is a great discussion of extraskeletal osteosarcoma.  Can the CT description and images of the mass be expanded so that the reader really knows there was no bone origin?  Thus far the ribs had not been described, so they cannot be ruled out as the origin.

Line 204:  The description of the CT is very important, as it is the best diagnostic tool to determine rib involvement / origin.

Line 228-229:  No omentum was moved to protect the lungs, please comment on this and utilize references.  The reader is interested in the pseudo pleura and it is very important to know if omentum is not necessary as it is for the use of polypropylene mesh.  This is new to veterinary medicine, so it is very important.

Line 248-250:  This statement cannot stand alone.  The follow up of the case portion MUST be expanded and images of the CT showing the complication must be included.  A full description of the literature associated with this complication following this type of reconstruction is the very reason for a publication such as this case.  Please expand the case follow up portion and then expand the discussion of titanium mesh in the chest.

Comments on the Quality of English Language

Minor changes as suggested

Author Response

Dear Reviewer

Thank you so much for giving opportunity to revise our manuscript “Chest Wall Reconstruction Using Titanium Mesh in a Dog with Huge Thoracic Extraskeletal Osteosarcoma” for MDPI in Animals. We want to extend our appreciation to you and the reviewers for taking the time and effort necessary to provide such insightful guidance. We have carefully considered comments offered by the reviewers. Herein, we explain how we revised the paper based on those comments and recommendations. We look forward to working further with you and the reviewers to move this manuscript closer to publication.

COMMENT 1.

Title:  Rather than use the colloquial term "huge" please use "large" or "very large" to describe the size of the mass.

RESPONSE 1.

Thanks for your detail comments. The revised content was reflected in the revised edition. We hope that these revised results will be a sufficient answer and will help to publish our research results in this journal.

Title : We changed “huge” to “large”.

COMMENT 2.

Line 44-46:  This sentence suggests that surgery improves survival time and is not palliative as it is in cases of appendicular osteosarcoma. The introduction or discussion should include this information with references.

RESPONSE 2.

Thank you so much for your careful review. This means that surgical treatment is recommended because non-surgical treatment is relatively associated with poor prognosis. And local tumor recurrence has a significant impact on the survival time.

We included this information with reference at introduction (line 45-47).

COMMENT 3.

Line 44-64:  The reader appreciates this paragraph. Can the authors provide information about using plates to support chest wall reconstructions when defects >4 ribs are made? This information does exist in the literature and is likely what readers will recall and to which they will refer.

RESPONSE 3.

Thanks for your detail comments. Here are the papers about using plate to support chest wall reconstructions when defects are more than 4 ribs. We included reference at line 51.

Reference 1) de Moya M, Nirula R, Biffl W. Rib fixation: Who, What, When?. Trauma Surgery & Acute Care Open 2017;2:e000059. doi:10.1136/tsaco-2016-000059

Reference 2) Tanaka H, Yukioka T, Yamaguti Y. Surgical stabilization of internal pneumatic stabilization? A prospective randomized study of management of severe flail chest patients. J Trauma 2002;52:727–32.

COMMENT 4.

Line 76:  Does the increase in ALP affect prognosis in cases like this as it does for appendicular osteosarcoma?  Please include this in the discussion of this manuscript.

RESPONSE 4.

Thank you so much for your careful review. Unfortunately, unlike appendicular osteosarcoma, there have been no studies reported on the association between elevated ALP levels and the prognosis of extraskeletal osteosarcoma. Further research is needed to explore this relationship.

We included this information at discussion with references.

COMMENT 5.

Line 80:  Please change "synechia" to "adhesions", as most readers understand that synechia refers to the eye.

RESPONSE 5.

Thanks for your detail comments.

Line 81 : We changed “synechia” to “adhesions”.

COMMENT 6.

Line 84:  Please change "and no metastasis" to "with no metastasis"

RESPONSE 6.

Thanks for your detail comments.

Line 85 : We changed “and no metastasis” to “with no metastasis”.

COMMENT 7.

Figure 1, part G:  Was there any change in the ribs associated with the mass?  Osteolysis, osteoproliferation?  Those changes are important in determining if this is axial osteosarcoma.

RESPONSE 7.

The initial CT showed invasion of the 8th rib. There were differences from typical rib-origin osteosarcoma on radiographs, and histopathology indicated a lack of association with the bone, suggesting it was an extraskeletal osteosarcoma. While the ideal surgical approach involves complete rib resection, not all affected rib segments were removed, which represents a limitation of this study.

COMMENT 8.

Line 100-101:  Thank you for including a very nice description of the intended surgical excision including ribs and cm to be removed and the method of preparing the mesh.

RESPONSE 8.

Thanks for your kind review.

COMMENT 9.

Line 110:  The instrument in Figure 2 part E seems to be a bone cutter or bone cutting forceps.  Please change "rongure" to one of the other terms.

RESPONSE 9.

Thanks for your detail comments.

Line 111 : We changed “ronguer” to “bone cutter”.

COMMENT 10.

Line 121:  If the mass did rupture, please address this in the discussion and how it may affect survival.  Please include any references as to the conclusions made.

RESPONSE 10.

Thanks for your detail comments. Tumor rupture can lead to hypovolemic shock, which may result in a poor prognosis. This is particularly concerning when the rupture occurs in the thoracic cavity, as it can cause respiratory distress, necessitating immediate intervention. However, there are currently no studies examining the relationship between the rupture of thoracic tumors and prognosis. Further research is needed.

We included this information at discussion with references.

COMMENT 11.

Line 138:  Please change "rongure" to "bone cutting forceps"

RESPONSE 11.

Thanks for your detail comments.

Line 138 : We changed “ronguer” to “bone cutter”.

COMMENT 12.

Line 141:  Please discuss how saline lavage minimizes the possibility of  metastasis.  If no literature is available to support this statement, please remove it form the portion on "seeding".

RESPONSE 12.

Thanks for your detail comments.

Line 142 : We deleted “to minimize the possibility of seeding”.

COMMENT 13.

Line 144:  Please change "tie" to "suture"

RESPONSE 13.

Thanks for your detail comments.

Line 144 : We changed “tie” to “suture”.

COMMENT 14.

Line 144-146:  Was no omentum used to protect the lungs from the mesh?  This is important and should be included in the discussion as well.

RESPONSE 14.

Thank you for your detailed comments. The omentum was not used to protect the lungs from the mesh. In many chest wall reconstruction studies using polypropylene or titanium mesh that I reviewed, the omentum was not utilized either. I was also concerned about potential lung irritation, but I found a study indicating that a pseudopleura forms after chest wall reconstruction, providing protection to the lungs. The details are included in lines 233-238: “Following chest wall reconstruction using a titanium mesh, a fibrous tissue layer called the pseudopleura forms within two weeks. The pseudopleura is a protective barrier against direct entry into the pleural cavity. However, the reasons for the formation of these layers remain unknown. This could be attributed to an inflammatory reaction secondary to the infection or the presence of the titanium mesh itself, which may have facilitated the formation of granulation tissue.”

COMMENT 15.

Line 155:  Please change "effaces" to "effaced".

RESPONSE 15.

Thanks for your detail comments.

Line 156 : We changed “effaces” to “effaced”.

COMMENT 16.

Line 153-157:  Were margins clear of neoplastic cells?  If yes, please provide the measurement of tumor-free tissue obtained.  This is very important to the reader and potentially to the complications of recurrence expected.

RESPONSE 16.

Thank you for your detailed comments. Unfortunately, although the tumor was resected with a 3 cm margin, the histopathological analysis revealed that the neoplastic tissue was only 0.2 mm from the nearest peripheral margin. Consequently, we are conducting serial CT follow-ups. 8 On CT images up to 8 months post-surgery, no clear progression of the disease has been observed.

COMMENT 17.

Figure 4:  Can images of the CT scan be included that are more appropriate for evaulation of lung for metastasis?  The current images do show the mesh very nicely.  But evaluation of bone, soft tissue, and lung are important for follow up.

RESPONSE 17.

Thanks for your detail comments. The evaluation of metastasis, particularly in the lungs, is typically more accurate when using different imaging views. However, in this case, no distinct signs of metastasis to bone, soft tissue, or other organs, including the lungs, were observed, so the images appeared normal, and thus were not included. If you could provide more details on which images you would like to add or what specific areas you are concerned about, it would be helpful for further adjustments.

COMMENT 18.

Line 190-206:  This is a great discussion of extraskeletal osteosarcoma.  Can the CT description and images of the mass be expanded so that the reader really knows there was no bone origin?  Thus far the ribs had not been described, so they cannot be ruled out as the origin.

RESPONSE 18.

Thanks for your detail comments. At the time of imaging, axial osteosarcoma could not be completely ruled out. The initial CT scan showed minor osteoproliferation in the 8th rib. However, no aggressive bone lesions, such as osteolysis typically seen in osteosarcoma, were observed in the initial or emergency CT and radiographs. The CT just before surgery showed dramatic osteogenesis of the inner mass, but minimal changes in the rib. Given the lack of bone association on histopathology and a history of an intramuscular mass extending into the thoracic wall, it is likely extraskeletal osteosarcoma.

COMMENT 19.

Line 204:  The description of the CT is very important, as it is the best diagnostic tool to determine rib involvement / origin.

RESPONSE 19.

Thanks for your detail comments. The CT scan findings indicated that the mass appeared to be invasive rather than originating from the rib.

COMMENT 20.

Line 228-229:  No omentum was moved to protect the lungs, please comment on this and utilize references. The reader is interested in the pseudo pleura and it is very important to know if omentum is not necessary as it is for the use of polypropylene mesh.  This is new to veterinary medicine, so it is very important.

RESPONSE 20.

Thank you for your detailed comments. In the many chest wall reconstruction studies using polypropylene or titanium mesh that I referenced, the omentum was not used. I was also concerned about potential lung irritation, but I found a study indicating that pseudopleura forms, which protects the lungs. Although the exact mechanism isn't clear, pseudopleura acts as a protective barrier, making the use of omentum unnecessary. This information is included in lines 233-238: “Following chest wall reconstruction using a titanium mesh...may have facilitated the formation of granulation tissue.”

COMMENT 21.

Line 248-250:  This statement cannot stand alone.  The follow up of the case portion MUST be expanded and images of the CT showing the complication must be included.  A full description of the literature associated with this complication following this type of reconstruction is the very reason for a publication such as this case.  Please expand the case follow up portion and then expand the discussion of titanium mesh in the chest.

RESPONSE 21.

Thank you so much for your careful review. In some cases, excessive rib retraction may lead to rib fractures, although these do not typically require corrective treatment.

We included CT image of fractured rib at Figure 4. and reference at the pharagraph.